# AdaKAN: Kolmogorov-Arnold Networks with Adaptive Spectral Decomposition for Time Series Forecasting

## Abstract

Real-world time series typically contain intertwined frequency components: low-frequency (trends), mid-frequency (periodicities), and high-frequency (short-term fluctuations or unexpected events), posing a significant challenge for accurate time series forecasting. To address this, we propose **AdaKAN**, a novel time–frequency Kolmogorov–Arnold Network (**KAN**) equipped with an **Ada**ptive Spectral Filter Module. Specifically: (i) AdaKAN adaptively decomposes time series into low-, mid-, and high-frequency components via learnable spectral thresholds. (ii) Fourier KAN captures global dependencies and periodic patterns, while Temporal KAN focuses on local structures and temporal dependencies. (iii) Each frequency band is processed with KANs of different orders with a soft weighting mechanism to better capture frequency-specific dynamics. Finally, time and frequency features are fused to form a comprehensive representation. Extensive experiments on multiple benchmarks demonstrate that AdaKAN consistently outperforms existing SOTA methods, offering a superior balance of accuracy and efficiency as an extremely lightweight architecture. Anonymous code is available in Appendix.

## 1 Introduction

The widespread deployment of IoT sensors and wearable devices results in the continuous and dynamic collection of time series data. Time series forecasting (TSF), which predicts future trends from historical observations, facilitates efficient planning and decision-making (Wu et al., 2024a). TSF has extensive applications across finance, meteorology, healthcare, and transportation fields (Huang et al., 2024; Wu et al., 2024b; Wang et al., 2024b; Shibo et al., 2025).

Current deep learning approaches for TSF primarily include CNNs, RNNs, MLPs, and Transformers. CNNs are effective at capturing local patterns but struggle with long-term dependencies (Zeng et al., 2024), while RNNs address temporal dependencies better but suffer from low efficiency and poor parallelization. Transformers such as iTransformer (Liu et al., 2024a) and PatchTST (Nie et al., 2023) have become mainstream due to their ability to model long-range dependencies via self-attention, yet their quadratic complexity and tendency to overfit small datasets pose key limitations (Wu et al., 2025). DLlinear (Zeng et al., 2023) has questioned their suitability for TSF, highlighting that permutation-invariant attention may impair temporal information preservation. While linears sometimes surprisingly outperform Transformers, they often fail under noisy or complex patterns. Recently, Kolmogorov–Arnold Network (KANs) have emerged as a promising alternative to MLPs in TSF (Xu et al., 2024a), offering efficient nonlinear representations with adaptive flexibility.

Time series typically exhibit non-stationarity and multi-periodicity, complicating the mapping from historical observations to future predictions. Recent TSF methods, such as Autoformer (Wu et al., 2021) and TimeMixer (Wang et al., 2024a), have integrated prior knowledge via seasonal-trend decomposition to improve performance. Frequency often offers complementary insights by revealing periodic structures and spectral distributions, critical for TSF (Jin et al., 2024), where patterns are often more distinct and interpretable (Eldele et al., 2024). Nevertheless, existing KAN-based research largely remains confined to time-domain modeling (Chen et al., 2025; Han et al., 2024a; Genet & Inzirillo, 2024). Even TimeKAN (Huang et al., 2025) employs FFT/IFFT techniques, it still operates KAN modules in the time domain, limiting their ability to capture localized spectral patterns.

Inspired by these insights, we propose AdaKAN, a lightweight **Ada**ptive frequency-aware **KAN** forecasting framework designed for complex and multi-frequency time series. Specifically, AdaKAN comprises four core components: (i) **Adaptive spectral filter module (ASFM):** employs adaptive low-pass, band-pass, and high-pass filters with learnable thresholding to decompose the input into low-, mid-, and high-frequency subseries. (e.g., ETTh1 in Figure 1); (ii) **FourierKAN:** models real and imaginary parts within each frequency band using separate KANs; (iii) **TemporalKAN:** applies inverse FFT followed by grouped convolutions and time-domain KANs

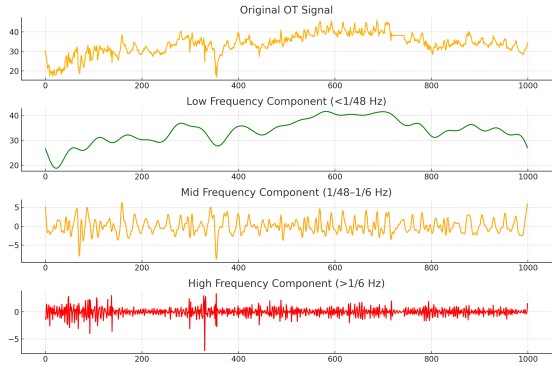

Figure 1: Decomposing 1000 samples of ETTh1 into low-, mid- and high- frequency components.

to extract local temporal features, also employing cross-attention to model interactions among frequencies. Notably, we introduce a soft-weighted multi-order mechanism in both FourierKAN and TemporalKAN, adaptively assigning orders based on input spectral features. (iv) **Fusion module:** integrates dual KANs outputs to comprehensively capture rich representations.

AdaKAN is lightweight, reducing complexity from $O(N^2)$ to $O(NlogN)$ employing the efficient Fast Fourier Transform (FFT). Although AdaKAN incorporates cross-attention mechanisms, they are restricted to frequency or local temporal interactions, avoiding the full quadratic cost of traditional attention, making it more efficient than self-attention (see Complexity analysis).

In summary, our contributions are as follows:

- We propose AdaKAN, a novel lightweight framework that integrates temporal-frequency KAN with an adaptive spectral filter module (ASFM) to effectively model and represent patterns across different frequencies by maximizing the flexibility of KAN.
- We revisit TSF from a frequency decomposition perspective and propose a hybrid framework that combines an adaptive spectral block and a dual-branch KAN. The spectral block uses Fourier transform with learnable thresholds to decompose inputs into low-, mid-, and high-frequency components, capturing trends, periodicity, and abrupt changes. The dual-branch KAN independently processes time- and frequency-domain features, leveraging their complementary strengths.
- Extensive experiments demonstrate that AdaKAN outperforms SOTA methods, validating its effectiveness and superiority in long-term TSF task.

## 2 RELATED WORK

**Kolmogorov–Arnold Network (KAN).** The Kolmogorov-Arnold theorem states that any multivariate function can be expressed by combinations of univariate functions and addition operations (Liu et al., 2024b). KAN replaces traditional linear weights with spline-based functions, enabling dynamic activation learning and improved interpretability. This flexibility makes KAN a compelling alternative to MLPs with fixed activation functions. KAN has been applied in diverse fields. U-KAN (Li et al., 2025) integrates KAN for efficient medical image segmentation. ViaSHAP (Alkhatib et al., 2025) learns Shapley value functions for direct prediction. GrokFormer (Ai et al., 2025) learns spectral filters via Graph Fourier KAN on classification task. Studies show KAN can surpass MLPs with fewer parameters (Vaca-Rubio et al., 2024). For TSF, T-KAN (Xu et al., 2024a) detects concept drift with symbolic regression. TimeKAN (Huang et al., 2025) integrates cascaded frequency decomposition with KANs to capture complex frequency patterns in the time domain. MMK (Han et al., 2024a) improves forecasting by assigning variables to KAN experts. Yet, most focus on the time domain, with limited use in the frequency domain. Given the multi-periodic and non-stationary nature of time series, we introduce three adaptive Fourier filters that dynamically divide the time series into high-frequency (sudden events), mid-frequency (seasonal cycles), and low-frequency (long-term trends) components, improving interpretability and robustness.

**Time Series Decomposition.** Real-world time series often consist of multiple latent patterns. To better capture these diverse dynamics, recent methods increasingly adopt decomposition strategies. N-BEATS (Oreshkin et al., 2020) and N-HiTS (Challu et al., 2023) incorporate decomposition into deep learning and achieve strong forecasting performance, but they overlook inter-channel dependencies critical to time series. DLinear (Zeng et al., 2023) uses moving averages to decouple trend and seasonality, while TimeMixer (Wang et al., 2024a) follows a fine-to-coarse strategy to separate sequences into multiple time spans. TimeKAN (Huang et al., 2025) proposes a decomposition learning block to accurately simulate complex patterns. Meanwhile, TimesNet (Wu et al., 2023) applies Fourier analysis to decompose sequences by learned periodic structures. Motivated by these advances, we propose a novel frequency decomposition method, employ adaptive low-pass, band-pass, and high-pass filters with learnable thresholds to divide the input sequence into low-, mid-, and high-frequency components, enabling precise modeling of frequency-specific dynamics.

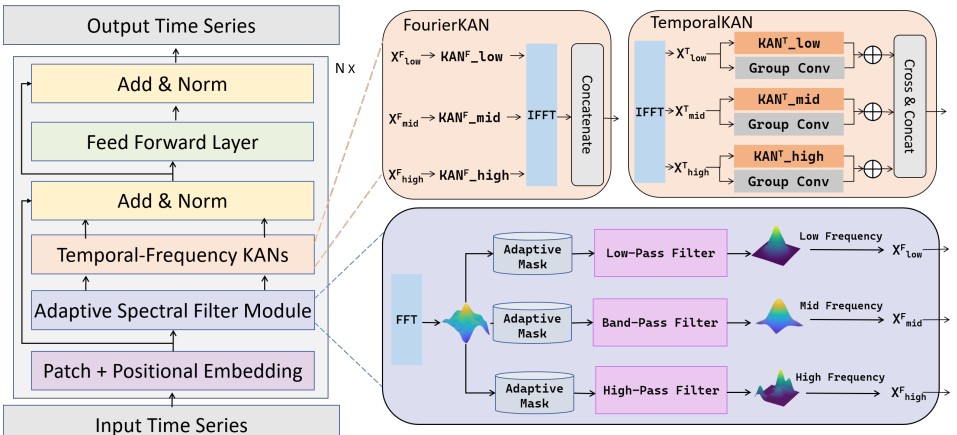

Figure 2: The structure of our proposed AdaKAN

# 3 METHOD

## 3.1 PRELIMINARIES

**Kolmogorov–Arnold Network (KAN).** The Kolmogorov–Arnold Representation Theorem (KART) serves as the mathematical foundation of KAN (Liu et al., 2024b), providing KAN with superior fitting capability and interpretability than MLP, which is grounded on the Universal Approximation Theorem (Kolmogorov, 1963), which states that for a function $f$:

$$f(x_1, \ldots, x_n) = \sum_{i=1}^{2n+1} \Phi_i \left( \sum_{j=1}^{n} \phi_{i,j}(x_j) \right) \qquad (1)$$

where $\phi_{i,j}$ are univariate functions that map each input variable $x_j$, and $\Phi_i$ are continuous functions. Each hidden neuron constructs an arbitrary function by composing multiple nonlinear transformations over the input features. For a single-layer KAN $\Phi$ with input dimension $n_{\text{in}}$ and output dimension $n_{\text{out}}$, the mapping is given by:

$$x_j^{\text{out}} = \sum_{i=1}^{n_{\text{in}}} \phi_{i,j} \left( x_i^{\text{in}} \right) \qquad (2)$$

where $x_i$ denotes the $i$-th dimension of $x$, and $\phi_{i,j}$ represents a learnable nonlinear function.

Based on the Kolmogorov-Arnold Representation Theorem, the architecture of KAN is designed as:

$$\text{KAN}(x) = \left( \Phi_{L-1} \circ \cdots \circ \Phi_1 \circ \Phi_0 \right) x \qquad (3)$$

where $\Phi_l$ is a layer of univariate functions Unlike traditional MLP (fixed activation functions), KAN innovatively trains learnable univariate activation functions $\phi_{i,j}$ using B-splines:

$$\phi(x) = w_b b(x) + w_s \, \text{spline}(x) \tag{4}$$

$$\text{spline}(x) = \sum_i c_i B_i(x) \tag{5}$$

where $w_b$, $w_s$ and $c_i$ are learnable weights, $B(x)$ is the basic function, and $\text{spline}(x)$ is parameterized as a linear combination of B-spline basis functions $B_i$.

## 3.2 OVERALL ARCHITECTURE

Our framework integrates two novel components in Figure 2: Adaptive Frequency Filter Block (ASFM) and Temporal-Frequency KAN, addressing the complexity of multi-frequency time series. ASFM decomposes the input into low-, mid-, and high-frequency components via learnable low-pass, band-pass, and high-pass filters with trainable thresholds, making it easier to learn valuable features of each frequency. After processing, these components are reconstructed into time-domain subseries using IFFT. FourierKAN models spectral characteristics within each frequency band, while TemporalKAN captures temporal dependencies using KANs with varying orders and group convolutions with different convolution kernels, allowing fine-grained extraction of local patterns. Together, they address both global and local dynamics, effectively learning trends, seasonality, and unexpected events for interpretable TSF.

## 3.3 EMBEDDING LAYER

Given an input time series $X \in R^{C \times L}$, where $C$ is the channel number and $L$ is the sequence length, $X$ is divided into $N$ non-overlapping patches $\{P_1, P_2, ..., P_N\}$, each of length $p$, totally patches $P_i \in R^{C*p}$. Each patch is mapped to a new dimension $d$, resulting in $P_i \to P_i' \in R^{C \times d}$. Position embeddings $E_i$ are added to each patch to preserve temporal order, giving $X_{PE_i} = P_i' + E_i$. The complete set of latent patches is $X_{PE} = X_{PE_1}, X_{PE_2}, ..., X_{PE_N}$, with the positional embeddings enhancing temporal correlation capture.

## 3.4 ADAPTIVE SPECTRAL FILTER MODULE

Time series are typically composed of multiple frequency components, and different frequency bands contain varying levels of dynamic information. Low frequencies reflect global trends, middle frequencies encompass periodic and medium-range state changes, and high frequencies comprise short-term fluctuations and noise interference. We model and learn the entire spectrum of the signal to decompose the whole frequency and independently learning each frequency can improve model robustness and generalization (Huang et al., 2025). To this end, we propose a frequency decomposition module based on learnable filters, which decomposes the original series $\mathbf{X}$ into low-frequency $\mathbf{X}_{low}^F$, middle-frequency $\mathbf{X}_{mid}^F$, and high-frequency $\mathbf{X}_{high}^F$. The entire structure is as follows.

**Fast Fourier Transformations.** For a given $X_{PE}$, its representation is calculated as:

$$
\begin{aligned}
X_F = \mathcal{F}[X_{PE}] &\in \mathcal{C}^{C \times L_F'} \\
&= \int_{-\infty}^{\infty} X_{PE} \cdot e^{-j2\pi fv} \, dv \\
&= \int_{-\infty}^{\infty} X_{PE} \cdot \cos(2\pi fv) \, dv - j \int_{-\infty}^{\infty} X_{PE} \cdot \sin(2\pi fv) \, dv
\end{aligned} \tag{6}
$$

Here, $\mathcal{F}[\cdot]$ denotes the 1D FFT operation, and $L_F'$ is the sequence length after Fourier transformation. $f$ denotes the frequency variable, $v$ denotes the integral variable, $j = \sqrt{-1}$ denotes the imaginary unit. We define three types of learnable filters in the frequency domain:

**Adaptive Low Pass Filter:**   Extract stable global trends helps with TSF.

$$\mathbf{X}_{low}^{F} = X_F \odot (|F| \leq \theta_{low}) \tag{7}$$

where $|F| \leq \theta_{low}$ represents a binary mask where frequencies below the threshold $\theta_{low}$ are retained, while others are filtered out.

**Adaptive Band Pass Filter:**   Capture the middle frequency to enhance model ability.

$$\mathbf{X}_{mid}^{F} = X_F \odot (\theta_{low} < |F| \leq \theta_{high}) \tag{8}$$

where $\theta_{low} < F \leq \theta_{high}$ denotes frequencies between the learnable thresholds $\theta_{low}$ and $\theta_{high}$ are retained, and others are discarded.

**Adaptive High Pass Filter:**   Retain small fluctuations can help improve responsiveness to changes.

$$\mathbf{X}_{high}^{F} = X_F \odot (|F| > \theta_{high}) \tag{9}$$

Similarly, $|F| > \theta_{high}$ means frequencies above $\theta_{high}$ retained, while others are discarded.

### 3.5 TEMPORAL-FREQUENCY KAN

**ChebyshevKAN.**   Vanilla KAN (Liu et al., 2024b) uses spline functions as the learnable univariate basic function $\phi$, but its complex recursive computation process hinders the efficiency of KAN. Thus, we instead adopt ChebyshevKAN to replace the B-spline function (SS et al., 2024) for its key advantages: (i) It can approximate smooth functions more efficiently with fewer parameters. (ii) Its inherent polynomial structure makes it particularly effective at capturing frequency patterns such as periodicity. ChebyshevKAN builds $\phi$ through a linear combination of Chebyshev polynomials of multiple orders. The Chebyshev polynomial is defined by:

$$T_k(x) = \cos(k \arccos(x)) \tag{10}$$

where $k$ is the highest order of the Chebyshev polynomials. Then, a single-layer ChebyshevKAN is:

$$x_j^{\text{out}} = \sum_{i=1}^{n_{\text{in}}} \phi_{i,j}\left(x_i^{\text{in}}\right) = \sum_{i=1}^{n_{in}} \sum_{k=0}^{K} \Theta_{i,k} T_k(\tanh(x_{in,i})) \tag{11}$$

Where $\Theta_{i,k}$ is the coefficient of $k$-th order Chebyshev polynomials acting on the $x_{in,i}$, and $\tanh$ is used to normalize the inputs to between $-1$ and $1$. By adjusting the highest order, we can control the fitting capability of KAN. This also inspires our design of the multi-order KAN to dynamically represent different frequencies.

**Multi-order with Soft Weighting.**   In KAN, order determines the nonlinear modeling capability of each channel (Huang et al., 2025). Adaptive selection of polynomial orders—e.g., assigning higher orders to high-frequency channels—can effectively enhance model performance and stability. We introduce a multi-order with soft weighting mechanism that assigns a learnable weight to each basis function order. Specifically, for each frequency band $x_{\text{low}}$, $x_{\text{mid}}$, and $x_{\text{high}}$, we assign a learnable weight vector $\boldsymbol{\alpha}^{(f)} = [\alpha_0^{(f)}, \alpha_1^{(f)}, \ldots, \alpha_n^{(f)}]$. The univariate $\phi(x)$ is computed as a weighted sum of basis functions, where the order weights are obtained via a softmax over trainable logits. This design adaptively emphasizes low- or high-order components, aligning model complexity with frequency hierarchy to enhance temporal pattern representation while avoiding overfitting and redundancy.

$$\phi^{(f)}(x) = \sum_{k=0}^{n} \alpha_k^{(f)} \cdot T_k(x) \tag{12}$$

where $f \in \{\text{low}, \text{mid}, \text{high}\}$, $\boldsymbol{\alpha}^{(f)} = \text{softmax}(\mathbf{w}^{(f)})$, $\mathbf{w}^{(f)} = [w_0^{(f)}, w_1^{(f)}, \ldots, w_n^{(f)}]$ is a set of learnable parameters. This mechanism enables the model to **automatically bias toward using a few polynomial orders** that best fit the corresponding frequency band.

**FourierKAN.** To handle complex-valued inputs $f \in X_{low}, X_{mid}, X_{high}$. we decompose it into real and imaginary parts $Re(X_f^F) = \int_{-\infty}^{\infty} X_{PE} \cdot \cos(2\pi f v) \, dv$ and $Im(X_f^F) = \int_{-\infty}^{\infty} X_{PE} \cdot \sin(2\pi f v) \, dv$. FourierKAN is composed of $\text{KAN}_{low}^F$, $\text{KAN}_{mid}^F$, $\text{KAN}_{high}^F$, and each owns a two-layer KAN; the network processes the real and imaginary components of $X_f^F$ separately. More specifically, the real part $Re(X_f^F)$ is firstly input into the two layer KAN to get the real output $Z_f^{Re} = \text{FourierKAN}(X_f^F)$. Then, the imaginary part $Im(X_f^F)$ is input into the network to get the imaginary output $Z_f^{Im} = \text{FourierKAN}(X_f^F)$. Due to the local plasticity of KAN, the features captured from the real and imaginary parts are integrated. Using the same KAN network for both parts ensures consistent feature learning and parameter sharing for meaningful signal reconstruction. Finally, the real and imaginary outputs are combined to form the final complex-valued representation:

$$\mathbf{Z}_f^F = \mathbf{Z}_f^{\text{Re}} + j \cdot \mathbf{Z}_f^{\text{Im}} \tag{13}$$

where $X_f^F \in \{X_{low}^F, X_{mid}^F, X_{high}^F\}$.

**TemporalKAN.** We first use IFFT ($\mathcal{F}^{-1}(\cdot)$) to transfer $X_{low}^F$, $X_{mid}^F$, $X_{high}^F$ back to the time domain: $\mathbf{X}_{\text{low}}^T = \mathcal{F}^{-1}(\mathbf{X}_{\text{low}}^F), \mathbf{X}_{\text{mid}}^T = \mathcal{F}^{-1}(\mathbf{X}_{\text{mid}}^F), \mathbf{X}_{\text{high}}^T = \mathcal{F}^{-1}(\mathbf{X}_{\text{high}}^F)$.

Similar to FourierKAN, TemporalKAN also has three KAN modules, $\text{KAN}_{low}^T$, $\text{KAN}_{mid}^T$, $\text{KAN}_{high}^T$. Each adopts a dual-branch parallel architecture to separately model temporal representation learning and temporal dependency learning in a frequency-specific way, using separate KANs to learn the representation and employing group convolution to capture the temporal dependency. Specifically, group convolution employs D groups of convolution kernels to perform independent convolution operations on the series of each channel. This allows the model to focus on capturing temporal patterns without interference from inter-channel relationships. For each input $X_f^T \in \{X_{low}^T, X_{mid}^T, X_{high}^T\}$, the process of group convolution and TemporalKAN is:

$$Z_{f,1}^T = \text{Conv}_{D \to D}(X_f^T, \text{group} = D) \tag{14}$$

$$Z_{f,2}^T = \text{TemporalKAN}(X_f^T) \tag{15}$$

Final output of TemporalKAN is the sum of the outputs from each KAN and group convolution:

$$Z_f^T = Z_{f,1}^T + Z_{f,2}^T \tag{16}$$

## 4 EXPERIMENTS

### 4.1 EXPERIMENTAL SETUP

**Datasets**. We evaluate AdaKAN on 8 benchmarks: 4 ETTs (ETTh1, ETTh2, ETTm1, ETTm2), Electricity (ECL), Exchange, Traffic, and Weather (Wu et al., 2021). Details are in Appendix.

**Baselines**. We assessed AdaKAN against ten baselines, including: 1) KANs: TimeKAN (Huang et al., 2025), MMK (Han et al., 2024b); 2) Mamba: DTMamba (Wu et al., 2024c). 3) Transformers: iTransformer (Liu et al., 2024a), PatchTST (Nie et al., 2023), Crossformer (Zhang & Yan, 2023); 4) MLPs: Dlinear (Zeng et al., 2023), TimeMixer (Wang et al., 2024a); 5) CNNs: TimesNet (Wu et al., 2023); 6) Frequency-based methods: FreTS (Yi et al., 2024).

**Implementation Details** are detailed in Appendix.

## 4.2 RESULTS

Table 1 compares AdaKAN with SOTA baselines across various datasets. AdaKAN consistently achieves strong performance, while iTransformer performs best on the Electricity (321 dim) and Traffic (862 dim) datasets due to its channel-wise self-attention, which effectively models dependencies in high-dimensional data. AdaKAN, TimeKAN, and TimeMixer also perform well on long-term forecasting, underscoring the effectiveness and generality of well-designed time series decomposition architecture. Unlike other baselines, AdaKAN introduces a novel adaptive frequency decomposition framework that learns hierarchical low-, mid-, and high-frequency components in both time and frequency domains, integrated with a multi-order KAN for superior long-term forecasting. Furthermore, our AdaKAN achieves 53 wins out of 80 tests under different metrics and conditions and maintains strong performance, underscoring its versatility and effectiveness.

Table 1: Multivariate long-term series forecasting results on input length 96 and prediction lengths $\in \{96, 192, 336, 720\}$. **Bold**: best, underlined: second best, $1^{st}$ Count: the number of the best results. Avg. means the average results from all four prediction lengths.

| Method | | AdaKAN (Ours) | | TimeKAN (2025) | | MMK (2024) | | TimeMixer (2024) | | DTMamba (2024) | | iTransformer (2024) | | PatchTST (2023) | | Crossformer (2023) | | TimesNet (2023) | | FreTS (2024) | | Dlinear (2023) | |
|---|---|---|---|---|---|---|---|---|---|---|---|---|---|---|---|---|---|---|---|---|---|---|---|
| Metrics | | MSE | MAE | MSE | MAE | MSE | MAE | MSE | MAE | MSE | MAE | MSE | MAE | MSE | MAE | MSE | MAE | MSE | MAE | MSE | MAE | MSE | MAE |
| ETTh1 | 96 | 0.374 | **0.393** | **0.367** | 0.395 | 0.374 | 0.397 | 0.386 | 0.400 | 0.386 | 0.399 | 0.386 | 0.405 | 0.414 | 0.419 | 0.423 | 0.448 | 0.389 | 0.412 | 0.402 | 0.416 | 0.386 | 0.400 |
| | 192 | **0.414** | 0.424 | **0.414** | **0.420** | 0.419 | 0.429 | 0.428 | 0.426 | 0.426 | 0.424 | 0.441 | 0.436 | 0.460 | 0.445 | 0.471 | 0.474 | 0.440 | 0.442 | 0.472 | 0.462 | 0.437 | 0.432 |
| | 336 | **0.443** | **0.432** | 0.445 | 0.435 | 0.461 | 0.450 | 0.489 | 0.454 | 0.480 | 0.450 | 0.487 | 0.458 | 0.501 | 0.466 | 0.570 | 0.546 | 0.495 | 0.470 | 0.518 | 0.484 | 0.481 | 0.459 |
| | 720 | 0.452 | **0.455** | **0.444** | 0.459 | 0.474 | 0.467 | 0.487 | 0.472 | 0.484 | 0.470 | 0.503 | 0.491 | 0.500 | 0.488 | 0.653 | 0.621 | 0.520 | 0.495 | 0.573 | 0.548 | 0.519 | 0.516 |
| | Avg. | 0.421 | **0.426** | **0.417** | 0.427 | 0.432 | 0.436 | 0.448 | 0.438 | 0.444 | 0.435 | 0.454 | 0.448 | 0.469 | 0.455 | 0.529 | 0.522 | 0.461 | 0.455 | 0.491 | 0.478 | 0.456 | 0.452 |
| ETTh2 | 96 | **0.285** | **0.334** | 0.290 | 0.340 | 0.301 | 0.353 | 0.303 | 0.351 | 0.290 | 0.340 | 0.297 | 0.349 | 0.302 | 0.348 | 0.745 | 0.584 | 0.332 | 0.370 | 0.347 | 0.399 | 0.333 | 0.387 |
| | 192 | **0.370** | **0.391** | 0.375 | 0.392 | 0.379 | 0.405 | 0.370 | 0.393 | 0.366 | 0.392 | 0.380 | 0.400 | 0.388 | 0.400 | 0.877 | 0.656 | 0.397 | 0.410 | 0.480 | 0.478 | 0.477 | 0.476 |
| | 336 | **0.356** | **0.395** | 0.423 | 0.435 | 0.432 | 0.446 | 0.401 | 0.422 | 0.380 | 0.409 | 0.428 | 0.432 | 0.426 | 0.433 | 1.043 | 0.731 | 0.446 | 0.448 | 0.519 | 0.509 | 0.594 | 0.541 |
| | 720 | **0.408** | **0.433** | 0.443 | 0.449 | 0.446 | 0.463 | 0.412 | 0.434 | 0.416 | 0.437 | 0.427 | 0.445 | 0.431 | 0.446 | 1.104 | 0.763 | 0.435 | 0.449 | 0.780 | 0.638 | 0.831 | 0.657 |
| | Avg. | **0.354** | **0.388** | 0.383 | 0.404 | 0.390 | 0.417 | 0.372 | 0.400 | 0.363 | 0.395 | 0.383 | 0.407 | 0.387 | 0.407 | 0.942 | 0.684 | 0.400 | 0.419 | 0.532 | 0.606 | 0.559 | 0.515 |
| ETTm1 | 96 | **0.312** | **0.355** | 0.322 | 0.361 | 0.320 | 0.358 | 0.321 | 0.361 | 0.325 | 0.360 | 0.334 | 0.368 | 0.329 | 0.367 | 0.404 | 0.426 | 0.347 | 0.383 | 0.352 | 0.385 | 0.345 | 0.372 |
| | 192 | **0.340** | **0.361** | 0.357 | 0.383 | 0.364 | 0.383 | 0.367 | 0.385 | 0.375 | 0.386 | 0.377 | 0.391 | 0.367 | 0.385 | 0.450 | 0.451 | 0.382 | 0.399 | 0.394 | 0.406 | 0.380 | 0.389 |
| | 336 | **0.371** | **0.386** | 0.382 | 0.401 | 0.395 | 0.405 | 0.391 | 0.403 | 0.396 | 0.405 | 0.426 | 0.420 | 0.399 | 0.410 | 0.532 | 0.515 | 0.409 | 0.420 | 0.430 | 0.431 | 0.413 | 0.413 |
| | 720 | **0.431** | **0.425** | 0.445 | 0.435 | 0.457 | 0.440 | 0.457 | 0.444 | 0.454 | 0.442 | 0.491 | 0.459 | 0.454 | 0.439 | 0.666 | 0.589 | 0.465 | 0.447 | 0.494 | 0.472 | 0.474 | 0.453 |
| | Avg. | **0.364** | **0.382** | 0.376 | 0.395 | 0.384 | 0.397 | 0.384 | 0.399 | 0.388 | 0.399 | 0.407 | 0.410 | 0.387 | 0.400 | 0.513 | 0.496 | 0.401 | 0.412 | 0.418 | 0.424 | 0.403 | 0.407 |
| ETTm2 | 96 | **0.169** | **0.253** | 0.174 | 0.255 | 0.176 | 0.261 | 0.176 | 0.258 | 0.177 | 0.259 | 0.180 | 0.264 | 0.175 | 0.259 | 0.287 | 0.366 | 0.189 | 0.266 | 0.194 | 0.290 | 0.193 | 0.292 |
| | 192 | **0.232** | **0.298** | 0.239 | 0.299 | 0.240 | 0.302 | 0.236 | 0.298 | 0.240 | 0.300 | 0.250 | 0.309 | 0.241 | 0.302 | 0.414 | 0.492 | 0.252 | 0.307 | 0.283 | 0.359 | 0.284 | 0.362 |
| | 336 | **0.277** | **0.335** | 0.301 | 0.340 | 0.299 | 0.342 | 0.298 | 0.338 | 0.310 | 0.345 | 0.311 | 0.348 | 0.305 | 0.343 | 0.597 | 0.542 | 0.321 | 0.349 | 0.360 | 0.407 | 0.369 | 0.427 |
| | 720 | **0.372** | **0.383** | 0.395 | 0.396 | 0.397 | 0.400 | 0.394 | 0.398 | 0.394 | 0.394 | 0.412 | 0.407 | 0.402 | 0.400 | 1.730 | 1.042 | 0.419 | 0.406 | 0.545 | 0.516 | 0.554 | 0.522 |
| | Avg. | **0.262** | **0.317** | 0.277 | 0.322 | 0.278 | 0.327 | 0.276 | 0.323 | 0.281 | 0.325 | 0.288 | 0.332 | 0.281 | 0.326 | 0.757 | 0.610 | 0.295 | 0.332 | 0.346 | 0.393 | 0.350 | 0.401 |
| Electricity | 96 | 0.164 | 0.261 | 0.174 | 0.266 | 0.166 | 0.256 | 0.156 | 0.247 | 0.166 | 0.256 | **0.148** | **0.240** | 0.181 | 0.270 | 0.219 | 0.314 | 0.171 | 0.274 | 0.189 | 0.276 | 0.197 | 0.282 |
| | 192 | 0.178 | 0.273 | 0.182 | 0.273 | 0.187 | 0.274 | 0.169 | 0.260 | 0.178 | 0.268 | **0.162** | **0.253** | 0.188 | 0.274 | 0.231 | 0.322 | 0.180 | 0.284 | 0.191 | 0.279 | 0.196 | 0.285 |
| | 336 | 0.192 | 0.282 | 0.197 | 0.286 | 0.204 | 0.290 | 0.186 | 0.277 | 0.197 | 0.289 | **0.178** | **0.269** | 0.204 | 0.293 | 0.246 | 0.337 | 0.200 | 0.300 | 0.206 | 0.296 | 0.209 | 0.301 |
| | 720 | 0.277 | 0.316 | 0.236 | 0.320 | 0.247 | 0.323 | 0.227 | 0.312 | 0.243 | 0.326 | **0.225** | 0.317 | 0.246 | 0.324 | 0.280 | 0.363 | 0.234 | 0.324 | 0.246 | 0.332 | 0.245 | 0.333 |
| | Avg. | 0.190 | 0.283 | 0.197 | 0.286 | 0.201 | 0.286 | 0.185 | 0.274 | 0.196 | 0.285 | **0.178** | **0.270** | 0.205 | 0.290 | 0.244 | 0.334 | 0.196 | 0.296 | 0.208 | 0.296 | 0.212 | 0.300 |
| Exchange | 96 | **0.083** | 0.203 | 0.086 | 0.205 | 0.096 | 0.219 | 0.091 | 0.215 | **0.083** | **0.201** | 0.088 | 0.206 | 0.088 | 0.205 | 0.256 | 0.367 | 0.122 | 0.255 | 0.094 | 0.222 | 0.088 | 0.218 |
| | 192 | **0.170** | **0.295** | 0.173 | 0.296 | 0.190 | 0.311 | 0.197 | 0.318 | 0.173 | 0.295 | 0.177 | 0.299 | 0.176 | 0.299 | 0.470 | 0.509 | 0.256 | 0.361 | 0.222 | 0.350 | 0.176 | 0.315 |
| | 336 | 0.342 | 0.424 | 0.366 | 0.438 | 0.383 | 0.447 | 0.416 | 0.472 | 0.346 | 0.427 | 0.331 | 0.417 | **0.301** | **0.397** | 1.268 | 0.883 | 0.432 | 0.479 | 0.431 | 0.492 | 0.313 | 0.427 |
| | 720 | **0.705** | **0.608** | 0.975 | 0.740 | 0.908 | 0.718 | 0.968 | 0.725 | 0.868 | 0.698 | 0.847 | 0.691 | 0.901 | 0.714 | 1.767 | 1.068 | 1.031 | 0.780 | 1.007 | 0.764 | 0.839 | 0.695 |
| | Avg. | **0.325** | **0.382** | 0.400 | 0.420 | 0.394 | 0.424 | 0.418 | 0.433 | 0.368 | 0.405 | 0.360 | 0.403 | 0.367 | 0.404 | 0.940 | 0.707 | 0.460 | 0.469 | 0.439 | 0.457 | 0.354 | 0.414 |
| Traffic | 96 | 0.527 | 0.338 | 0.575 | 0.368 | 0.511 | 0.324 | 0.462 | 0.295 | 0.487 | 0.317 | **0.395** | **0.268** | 0.466 | 0.298 | 0.522 | 0.290 | 0.591 | 0.315 | 0.557 | 0.329 | 0.650 | 0.396 |
| | 192 | 0.535 | 0.344 | 0.550 | 0.362 | 0.529 | 0.330 | 0.473 | 0.296 | 0.498 | 0.325 | **0.417** | **0.276** | 0.466 | 0.296 | 0.530 | 0.293 | 0.619 | 0.323 | 0.569 | 0.338 | 0.598 | 0.370 |
| | 336 | 0.541 | 0.349 | 0.576 | 0.378 | 0.545 | 0.334 | 0.498 | 0.296 | 0.511 | 0.334 | **0.433** | **0.283** | 0.482 | 0.304 | 0.558 | 0.305 | 0.637 | 0.339 | 0.566 | 0.337 | 0.605 | 0.373 |
| | 720 | 0.560 | 0.357 | 0.602 | 0.380 | 0.580 | 0.351 | 0.506 | 0.313 | 0.533 | 0.326 | **0.467** | **0.302** | 0.514 | 0.322 | 0.589 | 0.328 | 0.663 | 0.353 | 0.603 | 0.357 | 0.645 | 0.394 |
| | Avg. | 0.540 | 0.347 | 0.576 | 0.375 | 0.541 | 0.335 | 0.484 | 0.297 | 0.507 | 0.326 | **0.428** | **0.282** | 0.481 | 0.304 | 0.550 | 0.304 | 0.628 | 0.333 | 0.574 | 0.340 | 0.625 | 0.383 |
| Weather | 96 | **0.150** | **0.201** | 0.162 | 0.208 | 0.164 | 0.210 | 0.163 | 0.209 | 0.171 | 0.218 | 0.174 | 0.214 | 0.177 | 0.218 | 0.158 | 0.230 | 0.173 | 0.220 | 0.183 | 0.238 | 0.196 | 0.255 |
| | 192 | **0.192** | **0.241** | 0.207 | 0.249 | 0.210 | 0.251 | 0.211 | 0.254 | 0.220 | 0.257 | 0.221 | 0.254 | 0.225 | 0.259 | 0.206 | 0.277 | 0.219 | 0.260 | 0.251 | 0.312 | 0.237 | 0.296 |
| | 336 | **0.242** | **0.286** | 0.263 | 0.290 | 0.265 | 0.290 | 0.263 | 0.293 | 0.274 | 0.296 | 0.278 | 0.297 | 0.278 | 0.297 | 0.272 | 0.335 | 0.285 | 0.306 | 0.272 | 0.316 | 0.283 | 0.335 |
| | 720 | **0.312** | **0.338** | 0.338 | 0.340 | 0.343 | 0.342 | 0.344 | 0.348 | 0.349 | 0.346 | 0.358 | 0.349 | 0.354 | 0.348 | 0.398 | 0.418 | 0.376 | 0.365 | 0.349 | 0.377 | 0.345 | 0.381 |
| | Avg. | **0.224** | **0.266** | 0.242 | 0.272 | 0.246 | 0.273 | 0.245 | 0.276 | 0.254 | 0.279 | 0.258 | 0.278 | 0.259 | 0.281 | 0.259 | 0.315 | 0.263 | 0.288 | 0.264 | 0.311 | 0.265 | 0.317 |
| $1^{st}$ Count | | 53 | | 4 | | 0 | | 1 | | 1 | | 19 | | 2 | | 0 | | 0 | | 0 | | 0 | |

## 4.3 ABLATION STUDY

**Various Variants of AdaKAN.** To investigate the effectiveness of AdaKAN's various components, we conduct ablation studies on ETTh1, ETTm1, and Weather presented in Table 2. Removing group convolution and cross-attention significantly degrades performance, highlighting the importance of modeling temporal dependencies. Using fixed-order KANs at 2 yields inferior results compared to our multi-order with soft weighting design, confirming the effectiveness of gradually increasing the order to adapt to different frequency components. Among all components, removing ASFM causes the most substantial performance drop, with MSE increasing by 3.7%, 5.7%, and 5.9% on ETTh1, ETTm1, and Weather, demonstrating the value of our adaptive frequency decomposition. Moreover, eliminating either FourierKAN, TemporalKAN, or their fusion (T-F KAN) leads to consistent declines, MSE increased by 1.7%, 2.1%, and 3.0% on ETTh1, reflecting the importance of jointly modeling both spectral and temporal features. More details are in Appendix.

**KAN vs. MLP.** To investigate the contributions of KAN, we conduct ablation studies by replacing them with standard MLPs in different parts: (1) MLP_time and (2) MLP_freq: replace KAN with

Table 2: Ablation study of each component in AdaKAN.

| Variants | ETTh1 (avg.) | | ETTm1 (Avg.) | | Weather (avg.) | |
|---|---|---|---|---|---|---|
| | MSE | MAE | MSE | MAE | MSE | MAE |
| w/o Grop conv | 0.423 | 0.427 | 0.366 | 0.383 | 0.225 | 0.267 |
| w/o Cross-Att | 0.426 | 0.430 | 0.369 | 0.388 | 0.229 | 0.270 |
| w/o Multi-order | 0.424 | 0.429 | 0.367 | 0.385 | 0.228 | 0.268 |
| w/o ASFM | 0.437 | 0.439 | 0.386 | 0.397 | 0.238 | 0.277 |
| w/o FourierKAN | 0.429 | 0.432 | 0.372 | 0.391 | 0.230 | 0.271 |
| w/o TemporalKAN | 0.430 | 0.434 | 0.374 | 0.392 | 0.232 | 0.273 |
| w/o T-F KAN | 0.434 | 0.436 | 0.378 | 0.395 | 0.234 | 0.275 |
| AdaKAN | **0.421** | **0.426** | **0.364** | **0.382** | **0.224** | **0.266** |

MLP in the time and frequency domains, (3) MLP_time_freq: replace KANs in both domains. Table 3 reports the average results over four forecasting horizons $\in \{96, 192, 336, 720\}$ on ETTh1, ETTm1, and Weather. Across all settings, our multi-order KANs consistently outperform MLPs, highlighting their superior representation capacity and effectiveness as a compelling alternative.

Table 3: Ablation study by replacing KAN with MLP.

| Variants | ETTh1 (avg.) | | ETTm1 (avg.) | | Weather (avg.) | |
|---|---|---|---|---|---|---|
| | MSE | MAE | MSE | MAE | MSE | MAE |
| MLP_time | 0.425 | 0.429 | 0.368 | 0.385 | 0.227 | 0.269 |
| MLP_freq | 0.424 | 0.427 | 0.367 | 0.384 | 0.226 | 0.267 |
| MLP_time_freq | 0.427 | 0.430 | 0.371 | 0.388 | 0.230 | 0.271 |
| AdaKAN | **0.421** | **0.426** | **0.364** | **0.382** | **0.224** | **0.266** |

**Varying Look-back Window.** In TSF, extending the look-back window theoretically allows the model to access richer historical context, which may improve prediction accuracy. However, this benefit is not consistently observed in most Transformer-based models (Zeng et al., 2023). As shown in Figure 3, we vary the look-back window in $\{48, 96, 192, 336, 512, 720\}$, with the prediction horizon fixed at 96. Results show that iTransformer, PatchTST, and the CNN-based TSLANet fail to benefit from longer windows and even sometimes suffer performance degradation, revealing their limited temporal modeling capability. In contrast, the MSE of TimeKAN and our AdaKAN steadily decreases with longer windows, and AdaKAN consistently outperforms TimeKAN across all settings. This highlights AdaKAN's superior ability to capture both short- and long-term dependencies and to effectively extract information from extended historical sequences.

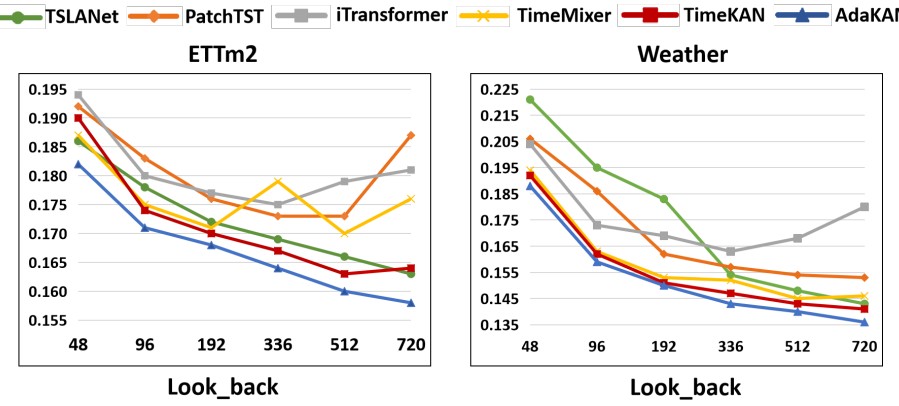

Figure 3: MSE on ETTm2 and Weather with varying look-back windows $\in \{48, 96, 192, 336, 512, 720\}$, and the prediction length is fixed to 96.

## 4.4 COMPLEXITY ANALYSIS

We evaluate the complexity of AdaKAN against other various TSF models-PatchTST, TimeKAN, Informer (Zhou et al., 2021), Fedformer (Zhou et al., 2022b), and FiLM (Zhou et al., 2022a) on Electricity, focusing on training parameters, MACs, and MSE under 96-step look-back and 720-step forecasting horizon, displayed in Figure 4. AdaKAN achieves superior predictive accuracy with substantially lower computational cost. While Transformer, Informer, Autoformer, FEDformer, and FiLM require 13.61M–20.68M parameters and 3.93G–5.97G MACs, their MSEs remain higher than AdaKAN. In contrast, AdaKAN matches the parameter scale of lightweight PatchTST and TimeKAN, yet reduces MACs by 85.94% and 80.64%, and improves MSE by 8.84% and 3.81%,

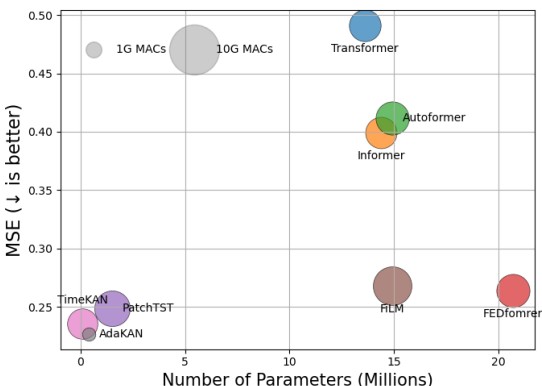

Figure 4: AdaKAN vs. baselines in terms of parameters and MACs count against MSE on Electricity with 96 look-back window and 720 forecasting length.

confirming AdaKAN's efficiency in TSF. These gains stem from AdaKAN's lightweight design: grouped convolutions, and the multi-order KAN structure. Thus, we cannot overlook that AdaKAN achieves excellent performance while requiring minimal computational overhead.

## 4.5 SENSITIVITY ANALYSIS

To obtain the optimal performance, we conducted a sensitivity analysis on ETTh1 and Weather, focusing on two key parameters: mask_ratio and dropout. As shown in Table 4, AdaKAN performs best when mask_rate is set to 0.25. A proper mask_rate value can ensure accurate learning representations while improving its robustness to adapt to unstable or data loss scenarios in reality. Dropout is used to prevent overfitting by randomly discarding a portion of neurons during training, making the network independent of any set of features. Table 5 displays that a dropout = 0.3 yields the best results on both ETTh1 and Weather. This indicates that appropriate dropout not only enhances generalization to unseen data, but also reduces the impact of noisy inputs and complex internal dependencies.

Table 4: Sensitivity experiments of mask_rate

| mask_rate | p=0.01 | p=0.1 | p=0.2 | p=0.25 | p=0.4 | p=0.5 |
|---|---|---|---|---|---|---|
| ETTh1 | 0.382 | 0.379 | 0.376 | **0.374** | 0.377 | 0.380 |
| Weather | 0.162 | 0.159 | 0.153 | **0.150** | 0.156 | 0.164 |

Table 5: Sensitivity experiments of dropout

| dropout | p=0.01 | p=0.1 | p=0.2 | p=0.3 | p=0.5 | p=0.6 |
|---|---|---|---|---|---|---|
| ETTh1 | 0.401 | 0.389 | 0.383 | **0.374** | 0.379 | 0.386 |
| Weather | 0.168 | 0.161 | 0.155 | **0.150** | 0.157 | 0.172 |

## 5 CONCLUSION

We present AdaKAN, a novel lightweight framework for TSF that seamlessly integrates dual KANs with an adaptive spectral decomposition module. AdaKAN employs three learnable filters—low-pass, band-pass, and high-pass—to adaptively decompose input sequences into low-, mid-, and high-frequency components based on trainable thresholds. To effectively learn and represent these specific components, we introduce a soft weighted multi-order mechanism in both Fourier KAN and Temporal KAN, which automatically learns weight assignments based on spectral features and further utilizes the high flexibility of KAN to capture long-term trends, periodic patterns, and short-term fluctuations in frequency and time domains. Extensive experiments on real-world datasets demonstrate that AdaKAN achieves SOTA forecasting performance and lightweight computational overhead. AdaKAN has opened up new avenues for TSF as a fundamental model.

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

## A RELATED WORK

### A.1 TIME SERIES FORECASTING

Time series forecasting aims to predict future values based on historical observations. Traditional methods such as VAR (Kilian & Lütkepohl, 2017) and ARIMA (Zhang, 2003) offer interpretability but often lack accuracy. In contrast, deep learning methods have gained traction for their ability to model nonlinear and complex dependencies, including CNNs, RNNs, Transformers, and MLPs (Lim & Zohren, 2021). CNNs, such as SCINet (Liu et al., 2022) focus on extracting local temporal patterns via convolution but struggle with long-range dependencies. RNNs like LSTM and GRU improve sequence modeling but are limited by sequential computation and slow training. Transformers like Informer (Zhou et al., 2021) and Crossformer (Zhang & Yan, 2023), have become dominant due to their global attention mechanisms, with variants like PatchTST (Nie et al., 2023) and iTransformer (Liu et al., 2024a) further enhancing performance. However, their high parameter counts and memory usage remain challenges. Recently, lightweight MLP-based models have shown superior accuracy and efficiency. For example, DLinear (Zeng et al., 2023) uses trend-seasonal decomposition, and FITS (Xu et al., 2024b) applies frequency-domain linear transformations. Building on this, we explore the potential of the KAN and propose an adaptive frequency decomposition module to better utilize frequency patterns for time series forecasting.

## B EXPERIMENTS

**Datasets**. To verify the model's performance, we evaluate the performance of our proposed Flames on 8 popular, well-established benchmarks: 4 ETT datasets (ETTh1, ETTh2, ETTm1, ETTm2), Electricity, Exchange, Traffic, and Weather. The datasets have been extensively adopted for benchmarking LSTF models and are publicly available on (Wu et al., 2021), covering domains such as electricity, transportation, energy, weather, and economy. Notably, we would like to highlight 3 large datasets: Electricity, Traffic, and Weather. They have more dimensions, so the results will be more stable and less prone to overfitting than other smaller datasets. (1) **ETT** [1] comprises two granularities collected from different regions in China, containing two hourly-level datasets (ETTh1, ETTh2) and two 15-minute-level datasets (ETTm1, ETTm2). Each dataset includes six power load features and a target "oil temperature" variable from July 2016 to July 2018. (2) **Electricity** tracks the power electricity consumption of 321 clients, converted to hourly measurements. This provides insights into usage patterns and enables demand forecasting, which is crucial for optimizing power generation and distribution. (3) **Exchange Rate** features daily exchange rates of different currencies from 8 different countries against the US dollar, allowing the model to predict currency fluctuations based on historical data. (4) **Traffic** records the road occupancy rate of different sensors on the San Francisco highway, including the traffic volume of 94 interstate westbound traffic in the Twin Cities metropolitan area, which can predict the traffic flow pattern, crucial to congestion management and urban planning. (5)**Weather**[2] contains 35,136 data points from a PV power station in northwest China, 15-minute intervals from January 1, 2020, to December 31, 2020. Besides, we will publicize this data as one benchmark. The statistics details are highlighted in Table 6.

**Implementation Details.** We conduct all experiments on an NVIDIA GeForce RTX 4090 Ti GPU with 64-bit Linux 5.15.0-56-generic, with 60/20/20 train/validation/test split for ETTs, and 70/10/20 for other datasets. Similar to (Zhou et al., 2023; Eldele et al., 2024) settings, we use the look-back window of 336 for ETTs, 96 for Exchange and Electricity, 512 for Traffic and Weather. All datasets are normalized during training (Kim et al., 2021). For baselines, we report their best results if setups match; otherwise, we rerun their code.

### B.1 MORE COMPARATIVE BASELINES

We compare our model with other common time series forecasting baselines, i.e. 1) Transformer-based: Autoformer (Wu et al., 2021) , FEDformer (Zhou et al., 2022b); 2) CNN-based: SCINet (Liu et al., 2022), MICN (Wang et al., 2023); 3) Linear-based: Rlinear (Li et al., 2023), TiDE(**?**);

---

[1]https://github.com/zhouhaoyi/ETDataset

[2]https://www.ncei.noaa.gov/data/local-climatological-data/

| Dataset | Dim | Dataset Size | Frequency | Domain |
|---|---|---|---|---|
| ETTh1 | 7 | (8545, 2881, 2881) | Hourly | Temerature |
| ETTh2 | 7 | (8545, 2881, 2881) | Hourly | Temerature |
| ETTm1 | 7 | (34465, 11521, 11521) | 15min | Temerature |
| ETTm2 | 7 | (34465, 11521, 11521) | 15min | Temerature |
| Exchange | 8 | (5120, 665, 1422) | Daily | Economy |
| Electricity | 321 | (18317, 2633, 5261) | Hourly | Electricity |
| Traffic | 862 | (12185, 1757, 3509) | Hourly | Transportation |
| Weather | 21 | (36792, 5271, 10540) | 10min | Weather |

Table 6: Dataset detailed descriptions. Dim denotes the variate number of each dataset. Frequency represents the sampling interval of time points. The dataset size is organized in (Train, Validation, Test).

4) Frequency-based: FiLM (Zhou et al., 2022a); 5) LLM-based: Tempo (Cao et al., 2024). Table 7 shows that our proposed Flames still exhibits significant advantages in all datasets and all lengths, For instance, on Weather, the MSE of our method decreases by 20.6%, 17.6%, 17.3%, 16.4%, 17.3%, 23.3%, 27.5%, and 33.7%. Similarly, the MAE exceeds other models by 15.8%, 8.6%, 16.9%, 17.1%, 8.3%, 26.7%, 26.1%, and 30.4%, which indicates the effectiveness and superiority of our framework in time series forecasting. Overall, our Affirm wins in 64 out of 80 results, and 8 second-best in the table.

| Method | | AdaKAN (Ours) | | Tempo 2024 | | Rlinear 2023 | | TiDE 2023 | | MICN 2023 | | FiLM 2022 | | SCINet 2022 | | FEDformer 2022 | | Autoformer 2021 | |
|---|---|---|---|---|---|---|---|---|---|---|---|---|---|---|---|---|---|---|---|
| Metrics | | MSE | MAE | MSE | MAE | MSE | MAE | MSE | MAE | MSE | MAE | MSE | MAE | MSE | MAE | MSE | MAE | MSE | MAE |
| ETTh1 | 96 | **0.374** | **0.393** | 0.400 | 0.406 | 0.386 | _0.395_ | 0.479 | 0.464 | 0.426 | 0.446 | 0.438 | 0.433 | 0.654 | 0.599 | _0.376_ | 0.419 | 0.449 | 0.459 |
| | 192 | **0.414** | _0.424_ | 0.426 | 0.421 | 0.437 | 0.424 | 0.525 | 0.492 | 0.454 | 0.464 | 0.494 | 0.466 | 0.719 | 0.631 | _0.420_ | 0.448 | 0.500 | 0.482 |
| | 336 | _0.443_ | **0.432** | 0.441 | 0.430 | 0.479 | _0.446_ | 0.565 | 0.515 | 0.493 | 0.487 | 0.547 | 0.495 | 0.778 | 0.659 | 0.459 | 0.465 | 0.521 | 0.496 |
| | 720 | _0.452_ | **0.455** | 0.443 | 0.451 | 0.481 | _0.470_ | 0.594 | 0.558 | 0.526 | 0.526 | 0.586 | 0.538 | 0.836 | 0.699 | 0.506 | 0.507 | 0.514 | 0.512 |
| | Avg. | **0.421** | **0.426** | _0.428_ | _0.427_ | 0.446 | 0.434 | 0.541 | 0.507 | 0.475 | 0.480 | 0.516 | 0.483 | 0.747 | 0.647 | 0.440 | 0.460 | 0.496 | 0.487 |
| ETTh2 | 96 | **0.285** | **0.334** | 0.301 | 0.353 | _0.288_ | _0.338_ | 0.400 | 0.440 | 0.372 | 0.424 | 0.322 | 0.364 | 0.707 | 0.621 | 0.358 | 0.397 | 0.346 | 0.388 |
| | 192 | _0.370_ | 0.391 | **0.355** | **0.389** | 0.374 | _0.390_ | 0.528 | 0.509 | 0.492 | 0.492 | 0.405 | 0.414 | 0.860 | 0.689 | 0.429 | 0.439 | 0.456 | 0.452 |
| | 336 | **0.356** | **0.395** | _0.379_ | _0.408_ | 0.415 | 0.426 | 0.643 | 0.571 | 0.607 | 0.555 | 0.435 | 0.445 | 1.000 | 0.744 | 0.496 | 0.487 | 0.482 | 0.486 |
| | 720 | **0.408** | **0.433** | _0.409_ | _0.440_ | 0.420 | 0.440 | 0.874 | 0.679 | 0.824 | 0.655 | 0.445 | 0.457 | 1.249 | 0.838 | 0.463 | 0.474 | 0.515 | 0.511 |
| | Avg. | **0.354** | **0.388** | _0.361_ | _0.398_ | 0.374 | 0.399 | 0.611 | 0.550 | 0.574 | 0.531 | 0.402 | 0.420 | 0.954 | 0.723 | 0.437 | 0.449 | 0.450 | 0.459 |
| ETTm1 | 96 | **0.312** | **0.355** | 0.438 | 0.424 | 0.355 | 0.376 | 0.364 | 0.387 | 0.365 | 0.387 | _0.353_ | _0.370_ | 0.418 | 0.438 | 0.379 | 0.419 | 0.505 | 0.475 |
| | 192 | **0.340** | **0.361** | 0.461 | 0.432 | 0.391 | 0.392 | 0.398 | 0.404 | 0.403 | 0.408 | _0.389_ | _0.387_ | 0.439 | 0.450 | 0.426 | 0.441 | 0.553 | 0.496 |
| | 336 | **0.371** | **0.386** | 0.515 | 0.467 | 0.424 | 0.415 | 0.428 | 0.425 | 0.436 | 0.431 | _0.421_ | _0.408_ | 0.490 | 0.485 | 0.445 | 0.459 | 0.621 | 0.537 |
| | 720 | **0.431** | **0.425** | 0.591 | 0.509 | 0.487 | 0.450 | 0.487 | 0.461 | 0.489 | 0.462 | _0.481_ | _0.441_ | 0.595 | 0.550 | 0.543 | 0.490 | 0.671 | 0.561 |
| | Avg. | **0.364** | **0.382** | 0.501 | 0.458 | 0.414 | 0.408 | 0.419 | 0.419 | 0.423 | 0.422 | _0.412_ | _0.402_ | 0.485 | 0.481 | 0.448 | 0.452 | 0.588 | 0.517 |
| ETTm2 | 96 | **0.169** | **0.253** | 0.185 | 0.267 | _0.182_ | _0.265_ | 0.207 | 0.305 | 0.197 | 0.296 | 0.183 | 0.266 | 0.286 | 0.377 | 0.203 | 0.287 | 0.255 | 0.339 |
| | 192 | **0.232** | **0.298** | _0.243_ | _0.304_ | 0.246 | 0.304 | 0.290 | 0.364 | 0.284 | 0.361 | 0.248 | 0.305 | 0.399 | 0.445 | 0.269 | 0.328 | 0.281 | 0.340 |
| | 336 | **0.277** | **0.335** | 0.309 | 0.345 | _0.307_ | _0.342_ | 0.377 | 0.422 | 0.381 | 0.429 | 0.309 | 0.343 | 0.637 | 0.591 | 0.325 | 0.366 | 0.399 | 0.372 |
| | 720 | **0.372** | **0.383** | _0.386_ | _0.395_ | 0.407 | 0.398 | 0.558 | 0.524 | 0.549 | 0.522 | 0.410 | 0.400 | 0.960 | 0.735 | 0.421 | 0.415 | 0.433 | 0.432 |
| | Avg. | **0.262** | **0.317** | _0.280_ | 0.328 | 0.286 | _0.327_ | 0.358 | 0.404 | 0.353 | 0.402 | 0.288 | 0.328 | 0.571 | 0.537 | 0.305 | 0.349 | 0.327 | 0.371 |
| Electricity | 96 | **0.164** | **0.261** | _0.178_ | 0.276 | 0.201 | 0.281 | 0.237 | 0.329 | 0.180 | 0.293 | 0.198 | _0.274_ | 0.247 | 0.345 | 0.193 | 0.308 | 0.201 | 0.317 |
| | 192 | **0.178** | **0.273** | 0.198 | 0.293 | 0.201 | 0.283 | 0.236 | 0.330 | 0.189 | 0.302 | 0.198 | _0.278_ | 0.257 | 0.355 | 0.201 | 0.315 | 0.222 | 0.334 |
| | 336 | **0.192** | **0.282** | 0.209 | 0.309 | 0.215 | _0.298_ | 0.249 | 0.344 | _0.198_ | 0.312 | 0.217 | 0.300 | 0.269 | 0.369 | 0.214 | 0.329 | 0.231 | 0.338 |
| | 720 | _0.227_ | **0.316** | 0.279 | 0.355 | 0.257 | _0.331_ | 0.284 | 0.373 | **0.217** | 0.330 | 0.278 | 0.356 | 0.299 | 0.390 | 0.254 | 0.355 | 0.254 | 0.361 |
| | Avg. | **0.190** | **0.283** | 0.216 | 0.308 | 0.219 | _0.298_ | 0.251 | 0.344 | _0.196_ | 0.309 | 0.223 | 0.302 | 0.268 | 0.365 | 0.214 | 0.327 | 0.227 | 0.338 |
| Exchange | 96 | **0.083** | **0.203** | 0.097 | 0.211 | 0.093 | _0.217_ | 0.094 | 0.218 | 0.097 | 0.228 | 0.098 | 0.256 | 0.267 | 0.396 | 0.148 | 0.278 | 0.197 | 0.323 |
| | 192 | **0.170** | **0.295** | 0.193 | _0.304_ | _0.184_ | 0.307 | 0.184 | 0.307 | 0.288 | 0.312 | 0.282 | 0.335 | 0.351 | 0.459 | 0.271 | 0.315 | 0.300 | 0.369 |
| | 336 | **0.342** | **0.424** | 0.362 | 0.438 | 0.351 | 0.432 | _0.349_ | 0.431 | 0.387 | 0.442 | 0.425 | 0.487 | 1.423 | 0.853 | 0.460 | _0.427_ | 0.509 | 0.524 |
| | 720 | **0.705** | **0.608** | _0.793_ | _0.689_ | 0.886 | 0.714 | 0.852 | 0.698 | 0.904 | 0.716 | 0.912 | 0.732 | 1.058 | 0.797 | 1.195 | 0.695 | 1.447 | 0.941 |
| | Avg. | **0.325** | **0.382** | _0.361_ | _0.411_ | 0.379 | 0.418 | 0.370 | 0.413 | 0.419 | 0.425 | 0.429 | 0.453 | 0.750 | 0.626 | 0.519 | 0.429 | 0.613 | 0.539 |
| Traffic | 96 | 0.527 | 0.338 | **0.476** | 0.343 | 0.649 | 0.389 | 0.805 | 0.493 | 0.519 | 0.309 | _0.514_ | **0.304** | 0.788 | 0.499 | 0.587 | 0.366 | 0.613 | 0.388 |
| | 192 | 0.535 | 0.344 | **0.496** | 0.355 | 0.601 | 0.366 | 0.756 | 0.474 | 0.537 | **0.315** | _0.528_ | 0.329 | 0.789 | 0.505 | 0.604 | 0.373 | 0.616 | 0.382 |
| | 336 | 0.541 | 0.349 | **0.503** | 0.356 | 0.609 | 0.369 | 0.762 | 0.477 | _0.534_ | **0.313** | 0.537 | _0.334_ | 0.797 | 0.508 | 0.621 | 0.383 | 0.622 | 0.337 |
| | 720 | _0.560_ | 0.357 | **0.538** | 0.376 | 0.647 | 0.387 | 0.719 | 0.449 | 0.577 | **0.325** | 0.622 | 0.363 | 0.841 | 0.523 | 0.626 | 0.382 | 0.660 | 0.408 |
| | Avg. | _0.540_ | 0.347 | **0.503** | 0.358 | 0.627 | 0.378 | 0.760 | 0.473 | 0.542 | **0.316** | 0.550 | _0.333_ | 0.804 | 0.509 | 0.610 | 0.376 | 0.628 | 0.379 |
| Weather | 96 | **0.150** | **0.201** | 0.211 | 0.254 | _0.192_ | _0.232_ | 0.202 | 0.261 | 0.198 | 0.261 | 0.195 | 0.236 | 0.221 | 0.306 | 0.217 | 0.296 | 0.266 | 0.336 |
| | 192 | **0.192** | **0.241** | 0.254 | 0.298 | 0.240 | _0.271_ | 0.242 | 0.298 | _0.239_ | 0.299 | 0.239 | 0.271 | 0.261 | 0.340 | 0.276 | 0.336 | 0.307 | 0.367 |
| | 336 | **0.242** | **0.286** | 0.292 | 0.332 | 0.292 | 0.307 | 0.287 | 0.335 | _0.285_ | 0.336 | 0.289 | _0.306_ | 0.309 | 0.378 | 0.339 | 0.380 | 0.359 | 0.395 |
| | 720 | **0.312** | **0.338** | 0.370 | 0.379 | 0.364 | 0.353 | _0.351_ | 0.386 | 0.351 | 0.388 | 0.360 | _0.351_ | 0.377 | 0.427 | 0.403 | 0.428 | 0.419 | 0.428 |
| | Avg. | **0.224** | **0.266** | 0.282 | 0.316 | 0.272 | 0.291 | 0.271 | 0.320 | _0.268_ | 0.321 | 0.271 | _0.290_ | 0.292 | 0.363 | 0.309 | 0.360 | 0.338 | 0.382 |
| $1^{st}$ Count | | 64 | | 10 | | 0 | | 0 | | 5 | | 1 | | 0 | | 0 | | 0 | |

Table 7: Multivariate long-term series forecasting results on the same input length=96, and different prediction lengths ∈ {96, 192, 336, 720}. A lower value indicates better performance. **Bold**: best, underlined: second best. $1^{st}$ Count: the number of the best results. Avg. means the average results from all four prediction lengths.

## B.2 Ablation Study

Ablation study of each component: (1) "w/o Group conv": remove Group convolution. (2) "w/o Cross-att": remove Cross-attention. (3) "w/o Soft-order": maintain a fixed low order 2. (4) "w/o ASFM": remove Adaptive Spectral Filter Module. (5) "w/o FourierKAN" and (6) "w/o TemporalKAN": remove FourierKAN and TemporalKAN. (7) "w/o T-F KAN": remove Temporal-Frequency KANs.

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
