# OpenReview forum: "AdaKAN: Kolmogorov-Arnold Networks with Adaptive Spectral Decomposition for Time Series Forecasting"
_ICLR.cc/2026/Conference — Submitted to ICLR 2026_

### Official Review · Reviewer_uHqX · 2025-10-31

**Soundness:** 2
**Presentation:** 2
**Contribution:** 2
**Rating:** 2
**Confidence:** 5

**Summary:**

The paper proposes AdaKAN, a dual-branch Kolmogorov–Arnold Network (KAN) framework for time-series forecasting. It introduces an Adaptive Spectral Filter Module (ASFM) that decomposes input sequences into low-, mid-, and high-frequency components using learnable thresholds in the Fourier domain. FourierKAN (for spectral modeling) and TemporalKAN (for local temporal dependencies)are combined via a fusion mechanism. Proposed model is evaluated on eight standard datasets, showing superior accuracy to transformer, MLP, CNN, and prior KAN baselines.

**Strengths:**

- This paper is well organized and easy to follow.
- Benchmark settings, datasets, and implementation details are clearly described.

**Weaknesses:**

- Limited novelty. The overall idea—decomposing a time series into low-, mid-, and high-frequency components and employing multi-order KANs to model frequency-specific dynamics—is almost identical to TimeKAN. The paper claims that TimeKAN only operates KAN modules in the time domain, yet TimeKAN already models frequency-domain signals with multi-order KANs after decomposition. The claimed contributions (“revisiting TSF from frequency decoupling,” “representing patterns across frequencies with KANs”) are essentially restatements of TimeKAN’s central ideas. Overall, the proposed framework extends prior work rather than introducing a fundamentally new principle.
- The core claimed novelty lies in the Adaptive Spectral Filter Module (ASFM) with learnable thresholds for frequency splitting. However, the paper did not explain how these thresholds are trained or optimized. The mask_ratio / mask_rate (should be consistent) hyperparameter discussed in Section 4.5 is never defined in the method, and its relationship with the spectral filter is entirely unspecified.
- Lack of validation for the learnable thresholds. Since the learnable thresholds are the only substantial difference from TimeKAN, the paper should provide concrete evidence that they matter, like 1) visualizations of learned thresholds across multiple datasets 2) convergence or stability curves during training, 3) ablations comparing fixed vs. learned thresholds
- The model applies multi-order KANs in both the time and frequency domains, whereas TimeKAN does so only in the frequency domain. This design should increase parameter count and MACs, yet Section 4.4 reports smaller numbers than TimeKAN without explanation.  More extensive and transparent evaluations on multiple datasets are needed to support any efficiency claims.
- About the presentation. Personally,  the paper should not spend excessive space on background like KAN or  FFT explanation. Besides,  Figure 2 is not self-contained and lacks clear indications of the signal flow, parameter sharing, and roles of each sub-module.

**Questions:**

The proposed framework appears conceptually very close to TimeKAN. Please authors clearly articulate what fundamental difference or new principle distinguishes AdaKAN from TimeKAN, and the difference is significant.

---

### Official Review · Reviewer_6Qu1 · 2025-11-01

**Soundness:** 3
**Presentation:** 2
**Contribution:** 2
**Rating:** 4
**Confidence:** 4

**Summary:**

This paper proposes a novel time–frequency Kolmogorov–Arnold Network (KAN) equipped with an Adaptive Spectral Filter Module. The adaptive spectral filter module adaptively decomposes time series into low-, mid-, and high-frequency components via learnable spectral thresholds. The proposed method mine the patterns of the time series in a Temporal-Frequency manner. Fourier KAN captures global dependencies and periodic patterns, while Temporal KAN focuses on local structures and temporal dependencies. Extensive experiments on multiple benchmarks demonstrate that the proposed approach consistently outperforms existing SOTA methods.

**Strengths:**

(1) The work demonstrates thorough experimental validation across multiple datasets and research questions.

(2) This study is grounded in solid theoretical foundations.

(3) The model architecture is clearly described.

**Weaknesses:**

(1) The selection of the comparative algorithm lacks justification.

(2) Lack of visualization.

(3) Self-inconsistent Appendix. All references appear twice in the list.

**Questions:**

(1) Decomposing a time series into low-, medium-, and high-frequency components inherently assumes that these components remain constant. This assumption contradicts the dynamic nature of time series, particularly in the context of time series forecasting tasks. What might be the author’s underlying intention?

(2) The main innovation of the paper lies in the integration of time series decomposition and KAN networks. Therefore, the comparative methods should be selected accordingly, rather than including other SOTA methods, even though the authors have compared numerous SOTA approaches.

(3) For the prediction performance, some visualization results can be added.

(4) Why is the ablation study conducted on only three datasets?

(5) KAN-based models for the time series forecasting have been proposed in the past. In addition, the proposed method also lacks theoretical insights and improvements.

---

### Official Review · Reviewer_jHNM · 2025-11-02

**Soundness:** 1
**Presentation:** 3
**Contribution:** 2
**Rating:** 4
**Confidence:** 4

**Summary:**

This paper proposes $\text{AdaKAN}$, a time-frequency $\text{Kolmogorov-Arnold Network}$ ($\text{KAN}$) for $\text{Time Series Forecasting}$ ($\text{TSF}$). It uses an $\text{Adaptive Spectral Filter Module}$ ($\text{ASFM}$) for frequency decomposition, processed by $\text{FourierKAN}$ and $\text{TemporalKAN}$ branches. The paper claims $\text{SOTA}$ results with $O(N \log N)$ complexity. My assessment is that serious technical flaws in the "adaptive" mechanisms undermine the paper's contribution and rigor. I recommend a Score 4 (Marginally Below).

**Strengths:**

The fusion of $\text{KAN}$ with frequency decomposition is a strong concept for non-stationary $\text{TSF}$. The architecture is efficient, reducing $\text{MACs}$ and parameters versus Transformers while maintaining competitive accuracy. The method segments input into low, mid, and high-frequency components, capturing trends, periodicity, and abrupt changes. Ablations confirm $\text{KAN}$ superiority over $\text{MLPs}$ and highlight the $\text{ASFM}$'s importance.

**Weaknesses:**

W1. Critical Soundness Issue in $\text{ASFM}$The $\text{ASFM}$ uses learnable thresholds $\theta_{low}$ and $\theta_{high}$ with non-differentiable binary masks for filtering:$$X_{low}^{F}=X_{F}\odot(|F|\le\theta_{low})$$The paper fails to explain how to backpropagate gradients for these thresholds. This omission undermines the technical soundness and "adaptive" claims of this core component.

W2. Limited Adaptivity in Multi-Order $\text{KAN}$The soft-weighted multi-order mechanism uses weights $\alpha^{(f)}$ derived from static logits $w^{(f)}$ for each frequency band $f$:$$\phi^{(f)}(x)=\sum_{k=0}^{n}\alpha_{k}^{(f)}\cdot T_{k}(x)$$Since $\alpha^{(f)}$ depends only on the band $f$, not the input, the weights are fixed globally per band. This prevents true per-sample adaptation and contradicts the "adaptive" claim.

W3. Insufficient Targeted Experimental ValidationExperiments fail to isolate the benefit of the adaptive filtering. A focused comparison against fixed frequency-domain models like $\text{FreTS}$ is missing. The $\text{ASFM}$'s superiority is thus unsubstantiated.

**Questions:**

Q1.Please detail the technical approach used to enable gradient propagation for the learnable thresholds $\theta_{low}$ and $\theta_{high}$ in the $\text{ASFM}$, given the non-differentiable binary masks.

Q2.If the multi-order weights $\alpha^{(f)}$ are truly adaptive, please provide the mathematical formulation showing the dependency of $\alpha^{(f)}$ on the input $X_{PE}$, as the current definition only depends on the fixed logits $w^{(f)}$.

Q3.Provide an ablation study comparing $\text{ChebyshevKAN}$ against $\text{Vanilla KAN}$ (using B-splines) to justify using the Chebyshev basis for this $\text{TSF}$ application.$\text{FreTS}$

Q4.Offer a focused comparative analysis against the $\text{FreTS}$ model to explicitly demonstrate how the $\text{ASFM}$'s adaptive capabilities yield concrete advantages over non-adaptive frequency-domain processing.

---

### Official Review · Reviewer_LAXq · 2025-11-08

**Soundness:** 3
**Presentation:** 3
**Contribution:** 2
**Rating:** 4
**Confidence:** 4

**Summary:**

This work investigates how to address the challenges posed by intertwined frequency components in time-series forecasting tasks. It proposes AdaKAN, a novel time–frequency Kolmogorov–Arnold Network equipped with an Adaptive Spectral Filter Module. AdaKAN introduces a learnable spectral filtering module that automatically splits signals into low-, mid-, and high-frequency bands, modeled respectively by frequency-domain and time-domain KAN branches with a multi-order soft-weighting mechanism. AdaKAN achieves state-of-the-art accuracy across multiple long-term forecasting benchmarks as an extremely lightweight architecture.

**Strengths:**

1.	This work proposes an Adaptive Spectral Decomposition Module (ASFM) that can dynamically split a time-series into low/mid/high frequency sub-bands, and sets the frequency-band thresholds as learnable parameters, thereby enabling the model to automatically adapt to the spectral structure of different datasets.

2.	This work introduces a time-domain – frequency-domain dual-branch KAN (Kolmogorov-Arnold Network). The frequency-domain branch (Fourier-KAN) models signals in the spectral domain, while the time-domain branch (Temporal-KAN) models in the time domain; additionally, a multi-order soft-weighting mechanism is introduced so that different frequency bands can select an appropriate nonlinear polynomial order, achieving a good balance between forecasting accuracy and efficiency.

3.	This work conducts comprehensive experiments on seven mainstream long-term time-series forecasting benchmarks, evaluating multiple forecasting horizons (96–720) and comparing against diverse baselines. The results consistently show that AdaKAN achieves state-of-the-art performance with notably strong robustness on long-horizon predictions, while ablation and efficiency analyses further verify the effectiveness of its adaptive spectral filtering, dual time–frequency branches, and low-complexity design.

**Weaknesses:**

1.	The authors present the adaptive spectral filter module (ASFM) as a major contribution of AdaKAN. However, adaptive spectral filtering in the frequency domain has already been explored in multiple prior works. This limits the novelty of ASFM [1].

2.	In line 83 the authors introduce AdaKAN’s contribution, including “We revisit TSF from a frequency decomposition perspective”. However, this point has already been claimed in the method TimeKAN [2]. The authors need to provide a more differentiated description to explain what the methodological contribution of AdaKAN is over existing work.

3.	The overall architecture’s motivation is not sufficiently clear. The authors lack explanation of the design principles behind AdaKAN’s framework. They do not explain why they needed both Temporal-Frequency KANs, or why Fourier-KAN and Temporal-KAN must be used complementarily.

4.	The core idea of AdaKAN is learning to split the time-series into frequency bands, but the experimental section lacks visualization or analytic experiments to demonstrate how the frequency-band learning helps prediction accuracy. The authors should consider including related experiments in the experimental section.

[1] Eldele, E., Ragab, M., Chen, Z., Wu, M., & Li, X. (2024). TSLANet: Rethinking Transformers for Time Series Representation Learning. In Proceedings of the 41st International Conference on Machine Learning (ICML 2024).
[2] Huang, S., Zhao, Z., Li, C., & Bai, L. (2025). TimeKAN: KAN-based Frequency Decomposition Learning Architecture for Long-term Time Series Forecasting. In Proceedings of the International Conference on Learning Representations (ICLR 2025).

**Questions:**

1.	In “Multi-order with Soft Weighting”, how do the learned polynomial orders behave after training? Can the learned order actually align with different frequency bands?

2.	The two hyperparameters in Section 4.5 are not mentioned earlier; the sensitivity analysis for these two hyperparameters is somewhat confusing. Could the authors explain the roles of these two hyperparameters?

---

### Meta-Review · Area_Chair_96Z8 · 2026-01-02

**Summary:**

The reviews are predominantly negative, specifically from 2, 4,4, and 5, with the most expressing dissatisfaction in Weakness parts. Consequently, LAXq
 has expressed concerns about limited novelty and contributions.
 jHNM concerns about the soundness issues; uHqX further shows concerns about limited novelty and poor presentation.All reviewers share common concerns about insufficient experiment comparison. Unfortunately, the authors do not answer such conerns, thus these substantial concerns remain addressed. Therefore, I persist in recommending the rejection of this paper in its current form.

**Reviewer Concerns:**

The authors have not uploaded the rebuttal, thus all concerns have not been solved.

**Reviewer Scores:**

All reviewers would keep or lower the score.

---

### Decision · Program_Chairs · 2026-01-26

Reject